# Nucleation and Growth-Controlled Facile Fabrication of Gold Nanoporous Structures for Highly Sensitive Surface-Enhanced Raman Spectroscopy Applications

**DOI:** 10.3390/nano11061463

**Published:** 2021-06-01

**Authors:** Eunji Lee, Sangwoo Ryu

**Affiliations:** Department of Advanced Materials Engineering, Kyonggi University, Suwon 16227, Korea; eg29560@kyonggi.ac.kr

**Keywords:** high-pressure thermal evaporation, nucleation and growth, gold nanoporous structure, surface-enhanced Raman spectroscopy, plasmonic nanostructure

## Abstract

The fabrication of porous metal structures usually involves complicated processes such as lithography or etching. In this study, a facile and clean method based on thermal evaporation at high pressure is introduced, by which a highly porous, black colored structure of Au can be formed through the control of homogeneous nucleation and growth during evaporation. The deposited films have different morphologies, from columnar to nanoporous structures, depending on the working pressure. These porous structures consist of Au nanoparticle aggregates, and a large number of nano-gaps are found among the nanoparticles. Thus, these structures indicate a much higher intensity of surface-enhanced Raman spectroscopy (SERS) when compared with commercial SERS substrates. The SERS intensity depends on the working pressure and thickness. Even circumstances that can induce agglomeration of nanoparticle aggregates do not deteriorate the sensitivity of SERS. These nanoporous structures based on high-pressure thermal evaporation are expected to provide a new platform for the development of low-cost and highly sensitive chemical sensors.

## 1. Introduction

Rapid and accurate probing of extremely small numbers of molecules has been a significant issue across several fields such as chemical sensors and pharmaceuticals. Recently, rapid and precise diagnostic tool kits have been developed for bacteria and viruses, and the COVID19 pandemic has boosted their significance. Surface-enhanced Raman spectroscopy (SERS) is one of the promising detection techniques because of its high sensitivity and simplicity of measurement [1,2,3,4,5,6,7,8].

SERS utilizes the plasmonic enhancement of the local electric field (referred to as localized surface plasmonic resonance (LSPR)) that occurs at nanoscale gaps of less than 10 nm, or “hot spots,” in various porous structures of gold or silver. These plasmonic nanostructures are known to amplify Raman signals by several orders of magnitude, which enables the probing of tiny amounts of target molecules [8,9,10,11,12,13,14,15,16,17]. The enhancement of the electric field becomes more noticeable as the size of the nano-gaps decreases with increasing density. Therefore, studies have focused on controlling the size and distribution of the nano-gaps [18,19,20,21,22,23,24,25,26].

One typical approach is the dispersion of metal nanoparticles synthesized using wet chemical processes [27,28,29,30,31,32,33,34]. Nanogaps formed around the connected nanoparticles act as hot spots that enhance Raman signals. Recently, Baek et al. reported that highly porous nanostructures obtained through sequential electrochemical anodization and reduction of gold thin films could provide a high density of hot spots, thereby enhancing the SERS activity [35]. These solution-based methods have advantages in terms of the large-scale production of SERS substrates. However, undesired chemical contaminants can be adsorbed onto plasmonic nanostructures during the wet process, which may suppress reproducibility.

Another approach is the cross-stacking of gold nanowire arrays through the nanotransfer printing method, as demonstrated by Jeong et al. [36]. In this case, LSPR occurs at the cross-points of the perpendicularly stacked nanowires, producing enhanced Raman signals. This method can control the size and distribution of the nano-gaps. However, the upscaling of nanotransfer imprinting remains challenging.

Therefore, commercial SERS substrates are being produced by dry etching of Si to fabricate columnar structures with nanosized openings among the columns and consecutive surface coating with gold or silver to induce LSPR [37,38,39]. While this manufacturing method utilizes well-established semiconductor process technologies, the entire sequence remains complicated, leading to high production costs. An alternative method developed by Park et al. still utilizes plasma etching of polymer films followed by surface coating with gold, which is not expected to reduce the production cost [40].

Herein, we introduce another approach to obtain highly sensitive SERS substrates that is clean and can be applied to large-scale production. In this work, a nanoporous gold structure was successfully fabricated by simple high-pressure thermal evaporation. It requires neither a complicated dry-etching process nor the deposition of gold under high vacuum. The nanoporous gold developed in this work consists of gold nanoparticle aggregates, and the LSPR arising from the hot spots among the aggregates induces a much-enhanced SERS activity compared to the commercial SERS substrate. In addition, it endures harsh environmental conditions such as thermal annealing. Owing to the isotropic evaporation of the source material, large SERS substrates can be obtained from a small amount of gold.

## 2. Materials and Methods

Nanoporous Au structures were fabricated on chilled Si wafers using a high-pressure thermal evaporation process. A detailed description of this process has been reported in previous studies [41,42]. Contrary to conventional evaporation methods that are carried out under high vacuum, the deposition chamber is filled with an inert gas to increase the working pressure up to a few Torr. In this study, an Au wire with a diameter of 1 mm was used as the evaporation source. An Au wire of 1–3 cm was placed on a tungsten boat, and the chamber was evacuated to 10^−3^ Torr. After filling the chamber with Ar to adjust the pressure, the Au wire was heated and evaporation was initiated. When the evaporation was completed, the power supplied to the evaporation source was quickly removed to prevent coalescence of the nanoparticle aggregates because of the residual heat. A shield installed between the source and the wafer also blocked the radiation heat from the source during evaporation. The effects of various control parameters, such as working pressure, evaporation time, deposition temperature, and annealing, on the evolution of morphology were explored.

Structural characterization was performed using X-ray diffractometry (Empyrean, Malvern Panalytical, Malvern, UK), scanning electron microscopy (JSM-7610F Plus, Jeol, Japan), and transmission electron microscopy (TEM; Tecnai F20, FEI Company, Hillsboro, OR, USA). The reflectance of the deposited nanostructure was measured using UV-Vis spectroscopy (Cary 5000, Agilent, Santa Clara, CA, USA).

For the surface-enhanced Raman spectroscopy measurement, a high-resolution Raman spectrometer (Jobin Yvon, Horiba, Japan) was utilized for rhodamine 6G (R6G) probe molecules diluted in de-ionized water. A 532-nm laser (0.5 mW) was irradiated, and the signal was accumulated for 10 s. Raman activity was measured for nanostructures with various surface morphologies, and the sensitivity was examined using different concentrations of R6G.

## 3. Results and Discussion

### 3.1. Nanoporous Structures of Au Synthesized through High-Pressure Thermal Evaporation

During thermal evaporation under a relatively high pressure, the evaporated metal atoms encounter repeated collisions with each other owing to the gas molecules while traversing through the deposition chamber. These collisions induce the formation of large clusters of evaporated metal atoms, whose surface mobilities are suppressed owing to their sizes as they arrive at the substrate surface. Finally, these clusters adhere to similar clusters that arrived earlier and do not diffuse, forming porous structures. In contrast, as the working pressure decreases, the evaporated atoms arrive at the substrate without collisions. These adatoms have a higher potential of diffusing on the surface, and, therefore, a denser structure should be formed. As indicated by the SEM analysis exhibited in Figure 1, the microstructure of Au thin films evolved from a nanoporous fractal-like structure (1 Torr, Figure 1a) to a relatively dense columnar structure (0.01 Torr, Figure 1c). The pores inside the Au thin film grown at approximately 1 Torr were so large that their porosity could not be measured using the Brunauer–Emmett–Teller (BET) method, which is typically used to measure the surface area and porosity of porous structures. The porosity estimated by the weight of the grown film and the dimensions obtained from the SEM analysis was higher than 90%, and it decreased as the working pressure was lowered.

The highly porous Au thin film is represented by black color in the inset of Figure 2a. Au thin films with relatively low porosity, which are grown under higher vacuum, are represented by greenish to yellowish color as the pressure decreased. Reflectance measurements verified that the highly porous Au thin film absorbs visible and near-IR spectra. The reason why the nanoporous Au thin film grown at 0.1 Torr has the lowest reflectance, close to zero, is that the optical properties of these nanoporous structures are affected by the thickness as well. While it was black, the grazing incidence X-ray diffraction (GIXRD) pattern presented in Figure 2b exactly matched that of pure gold, which verified that the obtained nanostructure was still that of gold.

The thickness of the nanostructured Au can be controlled by adjusting the evaporation time. As presented in Appendix A, the thickness decreases with decreasing evaporation time without changing the morphology at a certain pressure. Other variables that can affect the morphology of the obtained Au nanostructures are heat radiation from the evaporating source and deposition temperature. If the chamber is overpressurized, the increased number of gas molecules hinders the movement of the evaporated atoms, which lowers the deposition rate. The extended deposition time indicates that coalescence by heat radiation is more probable, which causes higher reflectance for visible light. To examine the effect of the deposition temperature, nanostructures formed on the chilled substrate and on the heated inside wall of the chamber were compared in Appendix A. The chilled substrate, whose temperature was maintained at about 100 °C during the evaporation, drove the formation of more porous structures when compared with the heated substrate.

As exhibited in Figure 1, the nanoporous Au formed at high pressure consists of aggregates of Au nanoparticles. All the individual nanoparticles of the aggregates were crystalline, as verified through the TEM analysis (Figure 2c). This supports the formation mechanism of the nanoporous structure: During evaporation, homogeneous nucleation and growth continues owing to repeated collisions, producing crystalline nanoparticles. These nanoparticles are weakly connected to each other in certain parts, whereas strong chemical bonds are also found. These aggregates result in the formation of nano-gaps of a few nanometers between the nanoparticles, which inspires the existence of plasmonic resonance.

### 3.2. Surface-Enhanced Raman Spectroscopy of Highly Porous Black Gold

#### 3.2.1. Enhancement of Raman Intensity by Nano-Gaps in Au Nanoparticle Aggregates

Nano-gaps formed among the Au nanoparticles enhanced the LSPR and increased the Raman intensity, as noted at the end of the preceding section. As indicated by the comparison presented in Figure 3a, the degree of enhancement measured by the peak intensity at 1361, 1505, and 1647 cm^−1^ increased as the pressure increased to 10^−1^ Torr. However, it began to decrease below 10^−1^ Torr. This is because the structure formed at lower pressure has a reduced number of nanogaps, whereas the structure formed at higher pressure also has low LSPR owing to the enlarged nanoparticle size. It should be noted that the Raman intensity for the nanoporous Au formed at 1 Torr is expected to be increased by reducing its thickness because the enhancement of the Raman intensity is dependent on the thickness of the nanoporous Au, as indicated in Figure 3b. In addition, the nanoporous Au had a sensitivity limit of 10^−6^ M for R6G molecules, as displayed in Figure 3c.

Compared to the commercial SERS substrate developed by Silmeco, which is fabricated by the dry etching of silicon wafer followed by the deposition of Au, the nanoporous Au investigated in this work exhibited much higher Raman intensity, as presented in Figure 3d. Considering the simple fabrication process, this difference is significant. The optimization of variables such as working pressure, thickness, and evaporation power would further increase the difference.

Another peculiar feature of the high-pressure thermal evaporation process employed in this study is that the surface is not homogeneously covered during the initial deposition, in contrast to conventional thin film deposition under high vacuum, where the deposited film homogeneously covers the substrate from the beginning. As indicated in Figure 4, both the surface coverage and thickness are evidently reduced with the decrease in the amount of the evaporation source. It should be noted that approximately 50% coverage of the nanoporous Au with a thickness of approximately 1 µm can remarkably enhance the SERS intensity (Figure 4c,e). Even a coverage of less than 50% results in a comparable SERS intensity (Figure 4b,e). This also accords prominence to this high-pressure thermal evaporation-induced nanoporous structure in terms of a cost-effective process for manufacturing nanostructures.

#### 3.2.2. Coalescence Still Maintains High SERS Sensitivity

One of the significant features of plasmonic nanostructures with high SERS activity is the existence of nano-gaps, which are formed among the Au nanoparticle aggregates in this study. This means that the SERS activity should be reduced if we suppress the formation of the nanogaps or remove the existing nano-gaps, as displayed in Figure 5a–c, where the Au nanostructures were annealed at 300 °C for 1 to 10 min to induce the coalescence of the Au nanoparticle aggregates. Contrary to expectations, however, all the annealed nanostructures exhibited comparable SERS activity regardless of the annealing time, as shown in Figure 5d. Even the sintered nanoparticles probably have sufficient nano-gaps for SERS. This promotes the applicability of nanoporous Au as a SERS substrate under various conditions that can significantly alter plasmonic nanostructures.

In addition to the annealing effect of the plasmonic nanostructures, the resistance of nanoporosity to liquid environment was also assessed. Agglomeration occurred in solvent such as water and ethanol causing the reduction of the nanoporosity. The degree of agglomeration was much higher in water than in ethanol. However, in hexane, the shape of nanoparticle aggregates and nanoporous morphology was still maintained. While the effects of SERS performance induced by liquid solvents were not investigated, both agglomerated and non-agglomerated nanostructures are expected to have comparable SERS sensitivity.

## 4. Conclusions

When physical vapor deposition, such as evaporation, is considered for the fabrication of thin films, it is usually understood that atoms move from the deposition source to the substrates without collision under high vacuum. By simply increasing the pressure, a highly porous structure can be synthesized through the homogeneous nucleation and growth of evaporated atoms. The obtained porous structures consist of nanoparticle aggregates with nano-gaps among the particles. These plasmonic nanostructures enhanced the Raman signal intensity as they were used for SERS, outperforming commercial SERS substrates. A simulation using the finite-difference time-domain (FDTD) method, which is under investigation, will elucidate the change of photophysical surface plasmonic resonance energy in terms of various morphologies and light absorption. With further studies on the optimization of the thickness of the deposited porous layer, the size of individual nanoparticles, porosity owing to pressure control, and high-pressure thermal-evaporation-based nanostructures explored in this study could provide a cost-effective alternative to chemical sensors.

## Figures and Tables

**Figure 1 nanomaterials-11-01463-f001:**
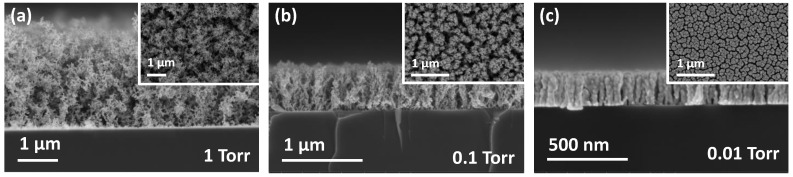
Porous structure of Au thin films obtained by nucleation and growth control during high-pressure thermal evaporation process at different pressure: (**a**) Fractal-like nanoporous Au thin film grown at 1 Torr; (**b**) intermediate nanoporous Au thin film grown at 0.1 Torr; (**c**) columnar Au thin film grown at 0.01 Torr.

**Figure 2 nanomaterials-11-01463-f002:**
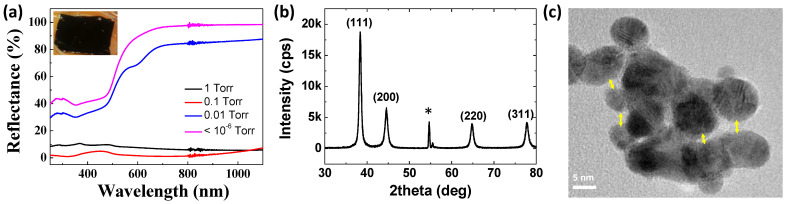
(**a**) Reflectance of Au thin film nanostructures grown at different pressure. Inset displays the digital image of the nanoporous ‘black gold’ grown at 1 Torr; (**b**) GIXRD analysis of the highly porous Au thin film grown at 1 Torr. All the reflections are labeled according to the JCPDS reference, and the star (*) indicates the reflections from the Si substrate; (**c**) high-resolution TEM image of the nanoparticle aggregate from nanoporous Au grown at 1 Torr. Lots of nano-gaps are found between the nanoparticles, as indicated by the yellow arrows.

**Figure 3 nanomaterials-11-01463-f003:**
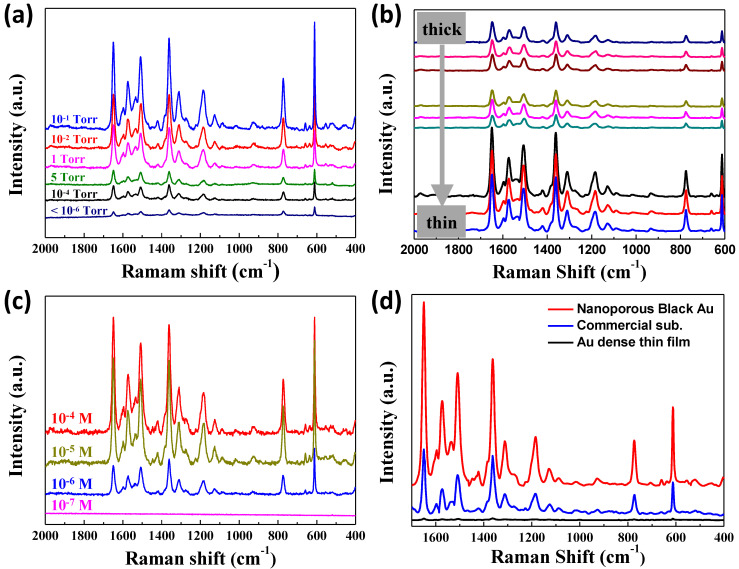
Results of SERS measurement: (**a**) Comparison of SERS spectra obtained from Au nanostructures grown at different pressure; (**b**) thickness dependence of SERS spectra for nanoporous Au grown at 1 Torr; (**c**) comparison of SERS sensitivity among various concentrations of R6G molecules; (**d**) comparison of SERS sensitivity of nanoporous Au to the commercial SERS substrate.

**Figure 4 nanomaterials-11-01463-f004:**
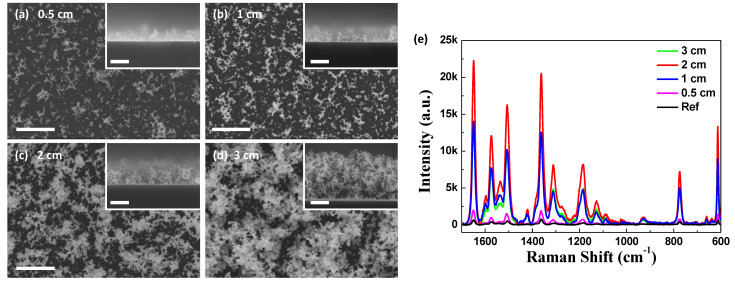
(**a**–**d**) Correlation between the thickness and the surface coverage of the nanoporous Au grown at 1 Torr. Thickness and surface coverage are controlled by different source amounts. All the scale bars are 500 nm; (**e**) SERS spectra from the nanoporous Au in (**a**–**d**). “Ref” indicates a spectrum from a dense thin film of Au.

**Figure 5 nanomaterials-11-01463-f005:**
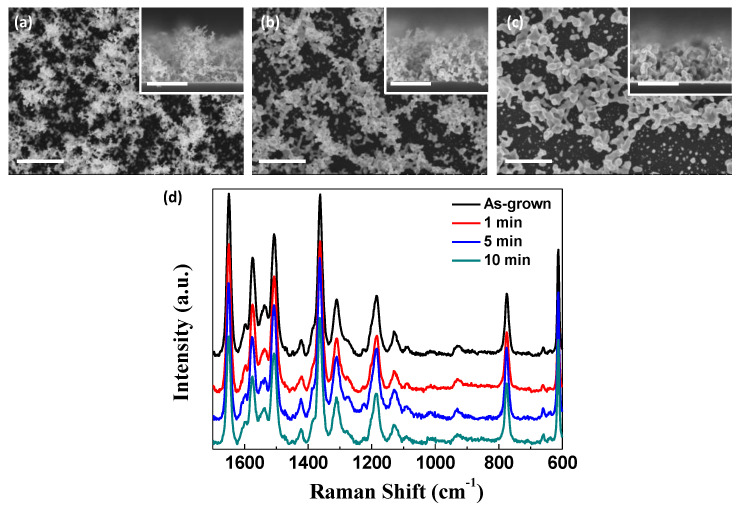
Effect of annealing of the nanoporous Au at 300 °C. Coalescence appears as the nanoporous Au is annealed for (**a**) 1 min, (**b**) 5 min, and (**c**) 10 min All the scale bars are 500 nm. (**d**) Comparison of SERS spectra from the annealed Au nanostructures with the non-annealed one.

## Data Availability

Not applicable.

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
