# Peer review of "Nucleation and Growth-Controlled Facile Fabrication of Gold Nanoporous Structures for Highly Sensitive Surface-Enhanced Raman Spectroscopy Applications"

_nanomaterials, 2021, doi:10.3390/nano11061463_

Round 1
Reviewer 1 Report
The paper "Nucleation and Growth-controlled Facile Fabrication of Gold Nanoporous
Structures for Highly Sensitive Surface-Enhanced Raman Spectroscopy
Applications" by Eunji Lee and Sangwoo Ryu iv very interesting and I strongly recommend its publication in this Journal.
The introduction paragraph well resume the different inherent approaches to the fabricartion of porous metal structures.
Methods and Results are clearly discussed, and the Conclusions are coherent with the researck.
I would only recommend to add few recent publications in the references.
Author Response
We thank reviewer 1 for the valuable comments. Based on the reviewer comments, we added recent publications in the references and revised the main text. The added references are yellow highlighted in the references as well as in the main text.
Reviewer 2 Report
Dear authors,
thank you for your interesting work about nanoporous gold layers for SERS applications.
In a goal of biological molecules detection and dosage, have you tried to graft some peptides or antibodies on trhe surface of your SERS detectors ? You mentionned the resistance of your layers to annealing protocols but have you assessed the resistance of nanoporosity to several aqueous but also non-aqueous solvents ?
Maybe a last question: have you an idea of the final temperature of the target at the end of evaporation process ? If yes, precise it in the text.
Sincerely yours
Author Response
We thank reviewer 2 for the interest in the possible application of our SERS substrates. Here, we provide a point-by-point response to the comments in blue.
Point 1. Have you tried to graft some peptides or antibodies on the surface of your SERS detectors?
Response) We are trying to graft some peptides or antibodies on the surface of our plasmonic nanostructures under the collaboration with biomedical research groups.
Point 2. Have you assessed the resistance of nanporosity to several aqueous but also non-aqueous solvents?
Response) Regarding the resistance of nanoporosity to liquids, agglomeration of the nanoparticle aggregates occurred in water and ethanol. The agglomeration rate was much higher in water. However, in hexane, the nanoporous morphology was still maintained. Therefore, we could say that the plasmonic nanostructures had resistance of the nanoporosity to non-polar solvents.
Point 3. Have you an idea of the final temperature of the target at the end of evaporation process?
Response) When we measured the temperature of the cooling block during the evaporation, it rose to about 100 °C with cooling so that coarsening could be prevented enough. We added this to the main text and yellow highlighted.
Reviewer 3 Report
The manuscript described the use of thermal evaporation technique to deposit gold nanoclusters on the substrate surface at a relatively higher pressure of 0.1 to 1.0 torr to observe columnar or nanoporous aggregation morphology structure. A brief description of nucleation and growth mechanism was given to allow the control of nanocluster formation during the residence time in the chamber. The manuscript is well-written and is publishable after additional comments provided regarding to photophysical surface plasmonic resonance energy generation properties in change of described “black gold” as compared with the regular gold nanoparticles well-known in the art, other than the enhanced absorption at NIR and IR wavelengths in Figures 2 and 3.
Author Response
We thank reviewer 3 for the essential comments. We are investigating the change of photophysical surface plasmonic resonance energy of our nanoporous gold in terms of the various morphologies obtained under different pressures, using simulations such as finite-difference time-domain (FDTD) method. It also includes comparison with the one of regular, monodisperse nanoparticles. We added this to the conclusion of the main text. In addition, as you mentioned, the enhanced absorption of NIR and IR wavelengths is the next step to expand our research, which can have other applications other than SERS. These new results will be incorporated in another paper.